# The Appropriation of Popular Culture: A Sensational Way of Practicing Evangelism of Korean Churches

**Min Hyoung Lee** 

Theology Department, Sung Kyul University, An Yang 14097, Korea; mhandhj@gmail.com

**Abstract:** Since the end of the 20th century, Korean churches have awakened to the fact that pop culture is enjoyed by a large segment of the population and thus provides a natural bridge between Christians and non-Christians. As a result, many Korean churches utilize popular cultural elements that Christians and non-Christians relish, such as movies, music videos, and images, as a way of demonstrating their evangelistic invitation to the world. They appropriate famous pop culture contents and present slightly modified materials through various channels, such as church websites, social media, and YouTube. This study focuses on the artistic technique of the evangelistic materials that Korean churches create. Based on the artistic understanding of appropriation, parody, and pastiche, this study examines whether the evangelistic imitations are "parodies" as they are introduced by their creators. I also look at ways another artistic style, "pastiches," might be more suitable to those appropriations than "parodies." Employing insights from the artistic analysis, this study explores which artistic style might be a better way of providing imitations of popular culture not so much as superficial entertainment but as a serious way to communicate the gospel. This will show how to appropriate popular culture as both faithful and efficient evangelistic methods.

**Keywords:** Korea; Christianity; evangelism; popular culture; appropriation; parody; pastiche

## 1. Introduction

When the Korean pop musician Psy swept the virtual and real world with his song titled "Gangnam Style," a Korean church made an imitation of his music video and uploaded it on YouTube.[1] Despite its flimsy quality, this video titled "Church Style" succeeded in engaging public attention.[2] In an interview, the pastor who produced "Church Style" and imitated Psy in the video said that he created a copy of Psy's "Gangnam Style" with an evangelistic purpose in his mind. He intended to produce a video affirming that the church is open to the world in the hope that people who watched the video would come to church (Yoo 2014).

The "Church Style" music video is only one example of a broader phenomenon of evangelistic imitations of popular culture created by Korean churches as a way of demonstrating their evangelistic invitation to the world by appropriating popular culture. In other words, Korean churches hybridize Christianity and popular culture for more appealing methods of communicating Christian messages. To understand this particular practice emerging in the Korean Christian context, it is essential to mention the "cultural mission" approach to evangelism that has had a powerful recent influence in

---

[1]  On 15 July 2012, Psy released "Gangnam Style" as the lead single of his sixth album and uploaded his music video on YouTube. Soon, it gained tremendous popularity and became the most-watched YouTube video. On 22 December 2012, his music video became the first YouTube video to reach one billion views. See Jung Joon Kim's article, "Gangnam Style: its Cultural Literacy and the Tasks of Christian Mission," in *K-Pop Wave and K-Christianity* (Kim 2013a).

[2]  According to *Kukmin Daily*, "Church Style" reached 110,000 views in its first week of release (Yoo 2014).

Korean Christianity. Sung Bin Yim, who is the chair of the Academy of Korean Cultural Mission, defines "cultural mission" as an evangelistic strategy with two primary presuppositions.[3] First, culture (and not merely individual persons) should be transformed by the Christian gospel or, to put it another way, culture is the object of Christian evangelism. Second, culture is a medium through which (or a tool by which) Christians practice evangelism. For the "cultural mission" approach, every realm of human culture must be transformed in accordance with the Christian gospel and thus realize the kingdom of God in this world (Yim 2011, p. 155).

The cultural mission strategy emerged in the early 1990s, but Korean churches have gone on to implement the cultural mission approach in a more robust way than Yim initially advocated. Some have come to regard cultural mission as a solution to the religious stagnation in Korean Christianity that Korean churches experienced in the late 20th century.[4] They construed the coming 21st century as an age in which culture-making would be far more critical than it was for Koreans in the decades immediately following the Korean War and the dramatic economic growth of Korean society and they anticipated the emerging importance of cultural materials, particularly popular culture (Kim 2014, pp. 273–84). Many Korean churches and their pastors have appropriated the cultural mission movement to deliver the gospel and invite non-Christians with an evangelistic means they understood to be in tune with the time, a time that accentuates the importance of culture (Kim 2008, pp. 375–80).

In a way that is parallel to the seeker-sensitive evangelistic approaches of mega-churches in the United States and elsewhere, popular culture became very important for Korean churches because it is enjoyed by a large segment of the population and thus provides a natural bridge between Christians and non-Christians. Korean churches began to bring popular culture into the church and utilize it for the sake of Christian evangelism, especially relying on such practices as the seeker-sensitive worship service and contemporary Christian music.[5] Both of these Christian practices utilize popular cultural elements that non-Christians relish, such as movies, plays, and popular music styles, so that persons would turn to and look favorably on churches.[6]

It is certainly true that insofar as an imitation directly presents slightly modified imitations of particular features of popular culture, it is a useful means of attracting people's attention. People tend to be drawn to cultural contents with which they are familiar (Park 2011, pp. 43–50). This results in various types of evangelistic replicas of contemporary mass media contents: Korean churches produce flyers promoting church events that imitate images of famous movie posters, sing popular songs slightly changing the lyrics to include religious terminology and references, and produce video clips that copy popular music videos, TV shows, or movies.[7] Despite the differences in the way

---

[3]　Cultural mission is also called as "Christian cultural movement" or "Cultural ministry." According to Yim, "Christian cultural movement" generally designates cultural missional ministry practiced outside the church, while "cultural ministry" describes ministerial practices using cultural contents within the church. Thus, Yim states, the concept of "cultural mission" encompasses both "Christian cultural movement" and "cultural ministry" (Yim 2011, pp. 154–59).

[4]　Meanwhile, some scholars interpret this time period from a different viewpoint. They think it was a time when the digital technology was rapidly advanced. At the end of 20th century, Korean society began to use Internet. Digital cameras came into wide use, and people could easily photoshop pictures downloaded from Internet. In the 21st century, people could freely watch videos through online video-sharing websites, and even they could produce visual images and upload them as UCC (User Created Contents) became popular. As the digital technology overwhelmed Korean society, Korean churches actively accepted and utilized it in various fields of ministry, such as worship services, management of communities, and production of promoting materials. Eventually, they could have an easy access to the materials of popular culture and utilize them to enrich their ministerial contents (Yoon 2011, pp. 207–8, 223–25).

[5]　In his article, Se Jong Park introduces using popular culture as a tool for evangelism. He states that a proper appropriation of popular culture can initiate communication between the church and the world. (Park 2011, pp. 39–40, 46).

[6]　In his book, *Eyes Wide Open*, William Romanowski pays attention to the fact that evangelical Christians frequently appropriate styles from popular culture for evangelistic purposes. He calls the productions of these evangelicals, such as CCM, as "confessional artworks" because they utilize popular cultural contents to approach non-Christians and deliver the gospel effectively (Romanowski 2007, pp. 30–32).

[7]　There are considerable numbers of the examples of the evangelistic parodies in Korean Christian context both online and offline. For example, when the movie *Avengers* was released in Korea, many churches made posters of "Amengers" that promoted church events by replacing the faces of the superheroes with the faces of their pastors or church members. Even when the Korean film noir, *New World*, hit the box office, a Korean church made a video clip that advertised its retreat by

each Korean Christian community takes advantage of popular culture, however, their evangelistic methods have one thing in common: they intentionally appropriate the contents of popular culture to create evangelistic media. Because of this dimension of imitation, moreover, those evangelistic means are regarded both by Korean churches themselves and by the Christian press reporting on them as products of artistic "parodies." This designation is not the result of an academic, artistic, or aesthetical examination of their styles, and there have been few academic studies clarifying the artistic term of reference of those evangelistic media. One plausible guess is that people perceive those evangelistic media as parodies simply because they are familiar with that particular term.

According to Marshall McLuhan, the signifier defines the signified. His well-known argument, "the medium is the message," suggests that content is inevitably determined by its form. Employing the insight from McLuhan's theory, each artistic genre determines the message of an artwork (McLuhan 1994, p. 25). Depending on what kind of artistic form, the content would have different meaning and function. If one knows the artistic forms of Korean churches' evangelistic imitations of popular culture, it will help him/her to understand why Korean churches utilize popular culture to practice evangelism culturally. It will also show eventually how the utilization of popular culture becomes helpful for Korean Christians to fulfill the purpose of evangelism, witnessing the gospel to the world.

Therefore, it is required to understand the exact meaning of parody and to review whether the definition of a parody is suitable to describe the artistic technique that Korean churches use when they utilize popular cultural contents for the sake of evangelism. If not, one should find a proper term of reference which fits the artistic meaning of Korean churches' contemporary evangelistic media. It is necessary not only for determining the adequate artistic description of Korean churches' evangelistic media but also for understanding the message that Korean churches ultimately deliver through those media.

My hypothesis is that what is really taking place is not parody but pastiche. Pastiche has something in common with parody in that both artistic styles use the technique of appropriation. Both borrow images or texts from the originals. Unlike parody, which copies the existing works with certain purposes, pastiche only aims at copying and pasting. Without any principles, it imitates arts recklessly and muddles up styles to create entertaining output (Jameson 1992, p. 9). If the resemblance between popular cultural contents and Korean churches' evangelistic media is the decisive factor in clarifying the artistic genre of those media, pastiche would be a term of reference which fits better than parody.

Therefore, to clarify the term of reference, I will explore the technique of appropriation which is the artistic foundation of both parody and pastiche. Based on the fundamental understanding of appropriation, I will go on to elaborate on the artistic meanings of parody and pastiche, focusing on their characteristics, similarities, and differences from each other. This discussion will provide a logical explanation which will provide an answer to the question as of why people usually mention every artwork using appropriation simply as parody. Ultimately, this article will study how a Christian theology of culture and evangelism might evaluate that evangelistic approach of imitating popular cultural contents. Based on theological understanding of Christian evangelism, it will examine whether the imitations of popular culture play a role as a significant evangelistic approach or not in the context of Korean Christianity. It will help to figure out how Korean churches understand the meaning of Christian evangelistic media and also provide a practical theological reflection on how to practice evangelism in relation to popular culture.

---

mimicking several scenes of the movie that contained violent lines and performances. A praise worship leader appeared in a singing competition TV show and sang several popular songs that he slightly changed some expressions of the original lyrics into church terms which Christians use only within Christian contexts. Furthermore, a Korean mega-church distributed an evangelistic flyer that copied a supermarket sweepstakes. Another mega-church created evangelistic material that contained only the faces of female members of the young adults group with an advertising copy of "do you want to have a girlfriend?" by parodying a famous matchmaking TV show.

## 2. Appropriation: The Fundamental Technique of Parody and Pastiche

To clarify the terms of reference which can describe the artistic genre of Korean churches' evangelistic imitations of popular culture in the most effective way, I will begin with exploring the fundamental artistic technique used in those media. Most artistic forms that contain imitating features share the methodology of appropriation. The term, appropriation, is a noun form of a verb, appropriate, which is originated from a Latin word, *propius*, which means "proper" or "property. The etymological meaning of appropriation is "to make something one's own" (Clifford 1990, p. 145). In the field of arts, it particularly means a technique that borrows pre-existing images from the arts, commercials, or mass media. Artists use the technique of appropriation by composing the original images with new images or simply cropping a part of them. They even rearrange those borrowed images like puzzles. Indeed, appropriation is an intentional artistic technique of imitation which primarily aims at copying the existing images or reconstructing the preformed artworks by making differences among similarities (Barnard 2009, pp. 6–10).

In fact, appropriation is an artistic skill which has developed throughout human history since the ancient times.[8] Despite its rich history that predates postmodern arts, however, appropriation recently came into its own as an artistic technique. It was after the end of the modernist society that art criticism began to pay attention to appropriation so that it became regarded as one of the typical features of postmodern art (Mun 2004, p. 317). There might be various reasons for the belated attention of art criticism to appropriation, but one of them is because it was neglected in the field of modern arts. After the Industrial Revolution, the world was gradually filled with newly invented technologies. In particular, the development of photograph technology threatened the artists of the time who had learned the technique of imitation as the essential art skill (Kim 2004, p. 58). Paintings that copied objects or nature as intact as possible became no better than pictures taken by photographers. As they sought a way for the survival of painting, modernist artists became skeptical about realistic art which emphasized the imitation of objects as the essential skill of painting.

As a result, artists intentionally avoided using the skills associated with realistic art, such as copy, imitation, or appropriation, in their art works. They treated appropriation as a technique of plagiarism and even made it taboo in the field of art (Mun 2004, p. 321). Instead, the emphasis of modern arts lay in the originality of a work of art. Artists developed techniques which could accentuate the originality and creativity that could not be expressed by copying or taking pictures of objects. From Impressionism to Abstract Expressionism, from the beginning to the end of modernism, various kinds of modernist artistic styles had developed, and none of them tolerated the similarity to pre-formed artworks. Each artistic movement was immersed in inventing creative artistic skills that could describe objects in a way that had never been existed previously.[9]

---

[8] The origin of appropriation tracks back to ancient times when Plato and Aristotle founded the theory of imitation. They understood that copy was not only an artistic technique but also a philosophical method through which human beings could express the substance of all creation. For them, objects were the visible reflections of the idea, the hypostasis of the universe, and arts, which was supposed to describe the objects by copying them, was a philosophical work itself. Thus, the value of a work of arts was estimated by how much it was similar to the object.Based on this understanding, imitation had developed as the primary method of the Western paintings in ancient times and throughout the Middle-Ages. In the Renaissance, the education of arts became systematized. The academia of arts emphasized the imitation of nature or the masterpieces as the core competencies for the artistic creation. This means that the tradition of the copy was maintained until the end of the Renaissance and the beginning of the Industrial Revolution. In other words, the tradition of the Western arts, which was represented by imitation, was shaken by the modernism. Unlike the artists of the previous ages, most of the modernism artists prefer expressing their inner world reflected on their subjective views to imitating classic masterpieces or objects. (Kim 2004, pp. 51–57).

[9] It does not mean that modern arts were completely freed from the legacy of imitation. Various modernist artists, such as Cezanne, Matisse, and Picasso, copied or reinterpreted masterpieces of the past (master) artists. Unlike the previous artistic technique of copying, which aimed at representing objects as they were, the modernist imitation method put more weights on deconstructing and reconstructing existing images. In other words, they secured their autonomy by transforming, not copying entirely, the masterpieces of the past. In this vein, their creations could be counted as a newly created works, but it is also true that they were still in the allegorical relation to the prior arts. (Newman 1989, pp. 99–100, 141).

Modern art, which was once characterized by the intention of the artists, their unique styles, and creative messages, finally reached the limit. Because of the skepticism of the art world about the originality, creativity, and purity of art, artists have accepted the reality that it was less possible to create a new artistic style since the 1960s. They recognized that the original art works were also only montages of existing images and forms. It became meaningless for them to discuss the originality and the meaning of an art piece as well as the intention of the creator (Barthes 1977, pp. 143–47). Within this artistic historical situation, postmodern artists, who challenged the limits of modernist artistic values, finally brought the techniques of copy and imitation back to the world of art. For them, the interpretation of the artists, who copied the originals, and the newly given intention reflected on the reproduction of the ready-made images were significant. They realized that the remaining way of creating art would be nothing but imitating "dead styles" of the past (Jameson 1992, pp. 17–18). It eventually gave validity to the contemporary appropriation technique.

Based on this historical background of art, appropriation becomes an artistic technique which is frequently used in postmodern art. Instead of creating unique and distinctive art which expresses the intrinsic value of objects, postmodern artists produce derivatives of pre-existing artworks by using appropriation. Artists collect images from various sources and edit them, and those collected images lose the initial meanings that the original creators assigned to them. Throughout this process, artists deconstruct the pre-formed artworks and replace them with new images and meanings. Even though the relationship between the genuine artworks and the new ones could be maintained, the newly created artworks present different meanings which reflect the contextual reality of where those artists live (Owens 1980, pp. 68–69). In this vein, appropriation is not just an artistic method which reconstructs the existing art pieces. Rather, it is a philosophical, political, and socio-cultural postmodern strategy which destructs vestiges of modernism that only value originality and creativity (Newman 1989, pp. 140–42).[10]

### 3. Parody and Pastiche: Two Possible Terms of Reference

Among different genres, parody and pastiche are representative styles of the arts applying the method of appropriation. Although both parody and pastiche utilize the same artistic method, the actual usage of these techniques varies considerably. Certainly, parody is a well-known term so that people choose to describe Korean churches' evangelistic media using popular culture. Meanwhile, pastiche is an unaccustomed word that only a few people know, and it has never been used to explain the same kind of evangelistic method. To figure out what exactly Korean churches do with the evangelistic replicas, therefore, it is important to understand the aesthetical meanings and functions of those art styles. In the following paragraphs, I will examine important distinctions between parody and pastiche, and eventually use that contrast to evaluate Korean churches' evangelistic appropriations of popular culture.

---

[10] One of the representative artworks of appropriation which show the postmodern deconstructive expression is Sherrie Levine's photograph, *After Edward Weston*. Levine creates this interesting piece by taking picture of Edward Weston's photograph in which Weston takes picture of his son. In Weston's picture, his son poses like a classic sculpture of a male nude. By taking a picture of his picture, Levine criticizes the originality of Weston's photograph. In other words, she conveys the message that Weston imitates the composition of the classic sculptures so that his picture is a product of appropriation. Thus, her photograph, which takes a picture of Weston's image, is an appropriated artwork in the same way that Weston's is. By borrowing an image from a copy of the classic artwork, she denies the concept of the original work. The technique of appropriation eventually enables Levine to demystify the modern arts' obsession over originality. At the same time, her photograph introduces that appropriation can break new ground in the field of postmodern arts.Rosalind Krauss calls Levine's technique as an "act of theft." However, she evaluates that Levine's artistic theft of images collapse the boundary between imitation and creation. Moreover, she calls Levine a postmodernist who showed that the hypothesis of modern art—the existence of the originality in arts—turned out to be fictitious. Her positive evaluation of Levine's photographs means that appropriation is eventually considered as an artistic technique of an era when it is barely possible to create original techniques and artworks (Krauss 1986, pp. 168–69).

### 3.1. Parody: The Known, Familiar, and Widely Used Term

#### 3.1.1. Definition of Parody

The term parody originates from a Greek compound word, *parodia*, which mixes a prefix, *para*, and a noun, *odos*. While the noun, *odos*, means a song, the prefix, *para*, has two different meanings. According to which meaning is applied, the entire meaning of *parodia*, or parody, is differentiated (Rose 1993, pp. 48–49). In general, the meaning of *para* is known as "counter," or "against." Thus, the primary purpose of parody is making contrasts between similar images or texts. Based on this conceptual definition, parody is known as a technique that aims at mocking a work of literature or artwork. In other words, parody is usually mentioned as a satirical artistic skill which compares the original texts or images to the ridiculously edited ones (Hutcheon 2000, p. 32). Artists create this kind of parody by appropriating a part of the pre-formed sources and completing the remainder with texts or images that they creatively produce.[11]

Meanwhile, the prefix, *para*, has another meaning of "beside." With this meaning of prefix, parody can be understood in a totally different nuance. Unlike the first concept, which emphasizes the contrasting feature of parody, this definition puts weight on the accordance or the similarity between the original and the newly created artwork. Instead of lampooning the original, parody becomes a means of "reconceptualizing" the texts or the images of the pre-formed sources (Hutcheon 2000, p. 33). Artists usually create this kind of parody to express their respect for the original authors or artists. Without harming the artistic value of the originals, artists reform those primary sources by reinterpreting and recontextualizing them. Given that a parody of the first meaning is close to satire, therefore, a parody of the second meaning is close to irony.[12] Like an irony which expresses the essence of an object in different ways, parody can shed new light on the art works of the past by modifying them depending on artists' preferences.[13]

#### 3.1.2. Artistic Functions of Parody

When it comes to considering both characteristics—mocking the genuine artwork with contempt and expressing one's respect for the preformed art piece—as essential purposes of parody, the definition of parody becomes wider than the concept that people usually have; parody is something more than a

---

[11] One of the representative artworks which reflects this concept of parody is Marcel Duchamp's *L.H.O.O.Q.*, which he drew in 1919. Duchamp drew a mustache and goatee on the postcard on which Leonardo da Vinci's *Mona Lisa* was printed. He also wrote initials of L, H, O, O, and Q at the bottom of the postcard. When it comes to pronouncing those initials in French, it is "Elle a chaud au cul," which means "she has a hot ass" (Huyssen 1986, p. 147). The purpose of the production of this scribble-like drawing with the raw initials, which became the title of this work, was not simply to mock da Vinci's masterpiece. Rather, Duchamp sneered at the bourgeois arts represented by the Louvre. He criticized the modernist idolatry of originality and creativity which was popular among the artists of the time who expressed the mindless adoration of *Mona Lisa* (Huyssen 1986, p. 148). Certainly, he drew a parody of *Mona Lisa* by creating the critical difference which was formed with the mustache, goatee, and the initials. Depending on the popularity of the genuine artwork, Duchamp eventually delivered his satirical message to the viewers of his parody.

[12] Although parody has satirical and ironic characteristics, it does not mean that those three artistic techniques are the same. Unlike parody, which needs the preformed texts or images as a part of its structure as well as its target, satire and irony do not need other materials except themselves. In other words, parody depends on the original materials to make a difference between the source and the parodied. Satire and irony, however, deliver the messages of the artists through the inconsistency between their internal and external meanings. Therefore, satire and irony are essential characteristics of parody, but they are not equal to parody (Rose 1993, pp. 81–82, 87–89).

[13] Various famous artists created art works related to this concept of parody. For example, Pablo Picasso's *Le Déjeuner sur l'herbe d'après Manet (Luncheon on the Grass, after Manet)* is a parody of Edouard Manet's *Le Déjeuner sur l'herbe (Luncheon on the Grass)* (McAuliffe 2014, p. 142). Although the brush of Picasso is different from Manet's style, the characters, their poses, and the color arrangement of Picasso's painting are similar to Manet's. Referring to Manet's work, Picasso presented a woman in the nude at the center of the painting, a man in black clothes who put his right hand on the grass, a woman crouched in the background, and the props on the grass. The only but remarkable difference is that Picasso simplified and planarized those characters so that he created his own style. Manet's classic painting inspired Picasso to appropriate its images, but Picasso created a reinterpreted parody of the masterpiece based on his respect for it.

caricature which only consists of satirical similarity. What eventually defines an artwork as a parody depends upon whether it has an irony that an artist creates by deforming the original images or not.

In her book, *A Theory of Parody*, Linda Hutcheon describes this irony as "repetition with critical distance, which marks difference rather than similarity" (Hutcheon 2000, p. 32). She explains that difference is the significant characteristic of parody which distinguishes parody from other artistic techniques of appropriation. According to her, parody makes differences by "trans-contextualizing" and "inverting" the contents of the parodied works. An artist edits the texts or images of the sources, or he/she can even completely change the structure of the pre-formed works (Hutcheon 2000, pp. 6–8). Depending on how he/she interprets the parodied works, the new work can have various kinds of differences. The critical differences created by an artist referring to the genuine works, which partially remain in the newly incorporated contents, present a singular contrast to the originals. People who discern those distinctions eventually understand the intention of the parodist reflected in his/her parody works.[14]

When it comes to regarding only its form, therefore, it seems that parody aims at a synthesis of the original and the newly created works. However, parody ultimately functions through "separation" and "contrast" of both works (Hutcheon 2000, p. 34). Texts or images reinterpreted and represented by a parodist play a role as a bridge which incorporates the parodied and the parody, but at the same time, those differentiated images deliver new meanings which are different from the messages of the original works. The trans-contextualized and inverted messages of the new works contain not only the marks of their sources but also critical distances from them. They critically deconstruct the originality of the materials and also creatively construct new meanings from those differences (Hutcheon 2002, p. 94).

Since parody delivers meaning through differences, parodists intentionally choose famous artworks as the sources for their parodies to help the viewers to recognize their purposes of production embodied in the differences. When an artist creates a parody of a masterpiece by editing it, people can identify the differences from the unknown parts of the revised masterpiece and finally, understand the message of the parodist. Certainly, it is a paradox of parody: parody aims at negating the myth of the original works, but cannot be identified as parody by the viewers without their prior knowledge of the originals.[15] Parody doubts and challenges the authority of the past, but its artistic value is only recognized by its origin in the past (Hutcheon 2000, p. 107). Thus, this paradox of parody should be read in this way: it does not mean that the parodied work loses its artistic value by being parodied. Instead, it gains new, different meanings in the world of the arts given by the artists of parody.

*3.2. Pastiche: The Unknown, Unfamiliar, and the Barely Used Term*

In general, people describe an art work as parody when they find its similarity to the pre-existing literature and art. Thus, Christian and non-Christian viewers and the journalists of the Christian press understand Korean churches' evangelistic media as parodies for the same reasons.[16] Certainly, it is hard to find any conjunctions with in-depth knowledge about parody from their conclusion. In other words, their evaluation of Korean churches' imitation of popular culture does not include any

---

14　Eventually, it results in the destruction of the authority of the original works. It also subverts the subject who interprets the arts from the creator to the viewer. In that parody challenges the originality of the arts, it is an artistic form which uses appropriation.

15　It is possible to state that this paradox of parody is related to the concept of defamiliarization found in the works of Russian formalists. They make the existing literary forms unfamiliar by substituting or subverting the accustomed rules of literature. The technique of defamiliarization is an inventive to Russian formalists who thought of art as a technical work which develops various artistic techniques that make viewers and readers tense. The defamiliarizing function of the paradox of parody is an appropriate technique for them to attract attention from the viewers and readers. (Park 2006, p. 260).

16　Almost every evangelistic imitation that Korean churches had created uploaded on social media, YouTube, or Christian newspapers. Interestingly, reporters who wrote articles about Korean churches' productions, people who wrote comments, as well as the creators of the imitations introduced those evangelistic media as "parodies." It is barely possible to find other designations.

discussions about parody's postmodernist nature, which challenges the modernist notion of originality and the importance of critical distance and difference.

Without serious analyses of what they do with the popular cultural contents, Korean Christians consider similarity as the conclusive evidence of parody and define their evangelistic tools as parodies. However, one possible refutation of this prevailing view which is deduced from the definition of parody reviewed in the previous paragraphs is that it is theoretically insufficient to define an art work as a parody only because it resembles its original materials. As repeated in the previous paragraphs, the meaning of artistic parody is more related to difference and distance than similarity (Hutcheon 2000, p. 6). My argument is that the proper term of reference for Korean churches' evangelistic media is another artistic genre, pastiche. In the following paragraphs, I will explore its meaning, comparing it with parody. I will also examine why pastiche fits more than parody as a description of what Korean churches create when using popular culture.

### 3.2.1. Definition of Pastiche

The term pastiche originates from an Italian word, *pasticcio*, which means a disparate mixture of materials (Rose 1993, p. 73). Its contemporary meaning is close to "scraping together" or "extracting things from various sources." In the field of postmodern arts especially, pastiche is practically used to describe an art form which appropriates images or texts from other sources and blends them together (Kim 1992, pp. 435–36). As mentioned above, pastiche is an artistic style that also uses the technique of appropriation. Unlike parody, which delivers the message of an artist through the repetitive differences from the original material, pastiche, also known as mixed imitation, solely aims at copying the original artwork partially or as a whole. The purpose of pastiche is pure and simple in that it does not have any critical intentions toward its sources. It places more weight on similarity than difference so that it does not maintain a certain distance from the sources (Park 2006, pp. 260–61).[17]

Pastiche came to the fore after Fredric Jameson mentioned it as one of the typical postmodern styles in his article, "Postmodernism and Consumer Society," in 1982. It is helpful to explore how he understood postmodernism first in order to understand why Jameson chose pastiche as the typical feature of postmodern society. According to Jameson, postmodernism is a holistic term that describes a socio-cultural status in which people experience a "radical break" with the older society (Jameson 1998, p. 19). Jameson sees a new type of capitalism—he calls it "late-capitalism"—that causes this radical change in society. The society of late-capitalism is dotted with an enormous range of changes in the political, economic, and cultural realms of humanity.[18] Although it would not

---

[17] An example of a pastiche which reflects the depthlessness and the historical discontinuity can be found in Jameson's comparison of Vincent van Gogh's *A Pair of Boots* and Andy Warhol's *Diamond Dust Shoes*. Jameson states that the old, worn-out, untied pair of boots in van Gogh's painting is a symbolic representation of reality. To interpret the meaning of this painting, he suggests, one needs to imagine the original context which moved van Gogh to draw this particular description of what he felt. According to Jameson, the initial situation surmised from van Gogh's painted clues is the "agricultural misery" (Jameson 1992, p. 7). The endless poverty, the intense labor, and the exhausted body of a peasant were the objects of van Gogh's painting and van Gogh condensed them into a pair of old boots. It shows that the signified is expressed through the signifier, and it also helps viewers to read the intention of the artist. Meanwhile, Warhol's work features various kinds of high heels in grayscale. Unlike van Gogh's artwork, it seems impossible to infer from where Warhol was inspired or why he created it from an image of a bunch of shoes. Jameson's evaluation of this unkind montage of the images of high heels from unknown sources is that it says nothing (Jameson 1992, p. 8). It rejects any kind of interpretation or even an attempt to interpret it. The superficial image of Warhol's work breaks the spatiotemporal connection between the past when he produced it and the present when a viewer sees it. By comparing one modern painting and one postmodern pastiche, Jameson certainly accentuates the characteristics of pastiche.

[18] He presents "new types of consumption; planned obsolescence; an ever more rapid rhythm of fashion and styling changes; the penetration of advertising, television and the media generally to a hitherto unparalleled degree throughout society; the replacement of the old tension between city and country, center and province, by the suburb and by universal standardization; the growth of the great networks of superhighways and the arrival of automobile culture" as some features of the postmodernist society. The ranges of those changes were too extensive and the appearances of those variations differed from context to context. It was impossible to handle every detail of this social shift. Accordingly, Jameson comprehensively defined postmodernism as the "cultural dominant." (Jameson 1992, p. 4).

be possible to encapsulate the appearances of those shifts in a word, they share the fundamental characteristic of postmodernism which is "depthlessness" (Jameson 1992, p. 17).

According to Jameson, this depthlessness, which is marked by superficiality and formality, is an indicator that distinguishes postmodern society from modern society (Jameson 1992, p. 9). People of modern society once had the dualistic depth models which made them think every object had essence and appearance, the inside and outside. The division of the inner and outer dimensions provides the logical background for the belief that inner truth is necessarily expressed by outer forms. Therefore, he describes the modern society as the age of hermeneutics in which people believed that one could measure the depth of an object by interpreting its appearance, the signifier. They thought it would lead them to the signified essence placed inside. However, as the world entered a new era, the age of postmodernity, people discounted the modernist idea of depth as metaphysical and unrealistic. The modernist understanding of depth was replaced by the concepts of "practices, discourses, and textual play (Jameson 1992, p. 13)." Sooner or later, Jameson states, the world became filled with the signifiers without the signified meanings, and the surfaces replaced the depth of the objective truth.

### 3.2.2. Artistic Functions of Pastiche

Postmodern society, in which the subjective individuals have disappeared, is filled with the absence of the unique styles and the depthless surfaces. In this context, it is impossible to invent any creative artistic styles. As Jameson states, the only possible way of (artistic) expression is nothing but imitating "dead styles" in a world in which creating a new style is impossible (Jameson 1998, p. 7). Therefore, he suggests pastiche, which extracts and combines meaningless signifiers, as the postmodern artistic form of expression. In particular, it is possible to understand pastiche as an art style which is influenced by the notions of intertextuality and "the death of the author" (Barthes 1977, pp. 143–44). As mentioned in the previous paragraphs, intertextuality describes a situation in which a text does not exist alone, but it is created under the influences of many other texts. In that an artwork of pastiche is created only by a patchwork of preforming images, it is an art of intertextuality. At the same time, pastiche deconstructs the concept of originality as it appropriates various images of the past arts. Since the meanings of those arts that the original artists assigned have become meaningless in the postmodern society, pastiche as a typical postmodern artistic style announces the death of the author through its ownerless images.[19]

In this vein, pastiche is distinguished from parody. Parody expresses criticism or respect for the arts of the past by maintaining a critical distance from those originals. However, pastiche does not have any intentions of criticism or satire: it solely copies and imitates existing images. Unlike parody which emphasizes difference, pastiche only has similarities to the sources. Therefore, Jameson names pastiche as "blank parody," which reveals the characteristic of pastiche: it only consists of the repetition of the appropriated images from which one cannot discover the intention of the original artists (Jameson 1992, p. 17). As mentioned above, it is impossible to figure out the intentions of the original authors from the mixture of random images. Instead, it only displays hybridity.[20] Since the notion of the authority of the original artists is extinguished with the end of modernism, one cannot trace the

---

[19]　Roland Barth describes this postmodernist challenge to the modernist norm of originality as "the death of the author." Based on this logic, the absence of the artist as a creator of the unique artwork results in the impairment of the authority of the original arts. Artworks are copied images, and therefore, parody is a reasonable artistic genre in the age when the authors are dead (Barthes 1977, pp. 143–47).

[20]　(Cultural) Hybridity is a frequently used term in the studies of post-colonialism. It depicts the cultural status that there is no single culture: every culture is intermingled with other cultures. In the postmodernist society, it is impossible to find an independent cultural form which has never been affected by other cultural sources. Every culture creates ambivalent, or even multivalent, cultural contents by blending various cultural elements. Hybridity designates this new cultural wave (Burke 2009, p. 51).

original artworks from the mingled images of pastiche. Therefore, pastiche is only a combination of images which have lost their historical links.[21]

Jameson thinks the discontinuity of the historical link of pastiche is a mark that reveals the "schizophrenic" aspect of postmodernist culture (Jameson 1992, p. 28). From the perspective of linguistics, pastiche is a mixture of signifiers which lose their signified meanings. It means that the meaning of an artwork of the past is not continued in the current media. The schizophrenic breakdown isolates images from their meanings assigned throughout history. Insofar as it is consisting of those isolated images, pastiche cannot convey any messages to its viewers. Since it does not provide any clues related to the meanings of those signifiers, viewers cannot guess what the pasticheur tries to say through his/her creation. Jameson sees pastiche as an insufficient medium for an artistic expression of ideological messages. Therefore, the products of pastiche cannot be used as a means of exposing reality or criticizing contemporary socio-cultural issues (Won 2010, pp. 428–30).

## 4. What Korean Churches Do with Popular Cultural Contents for the Sake of Evangelism

As examined in the previous paragraphs, the aesthetical meanings of parody and pastiche are much more complicated than the conventional understanding of them. What people usually mention as parodies are not always parodies; in fact, some are parodies, while others are pastiches. The conceptual confusion and incorrect usage of those terms gets worse when people associate parody with similarity. Even though parody is less related to similarity than pastiche, the prevalent view reminds people of the term parody whenever they face new artworks which are similar to the old ones. Certainly, the development of Korean evangelistic media imitating popular culture happened through a similar train of logic. People who made, saw, and wrote about those media found familiarity and similarity first, and spontaneously recalled the P-word, parody.

Korean churches produced evangelistic media, such as the "Church Style" video clip, by appropriating popular cultural content of the time. The evidence of the fact that they appropriated popular culture is the similarity of those media to the popular content. Unless one is isolated from popular cultural media, such as TV, Radio, or the Internet, most people easily recognize the sources of those evangelistic media. As the "Church Style" video clip reminds its viewers of the worldly famous music video, "Gangnam Style," people who see the imitation movie posters produced by Korean churches can tell each title of the original films. The ones who revealed that several evangelistic methods of Korean Christianity imitated the provocative contents of popular culture were not only Korean churches which created those media, but also people who saw them. What made these conclusions possible was the similarities of those evangelistic media to the pre-formed popular cultural content. The similarity eventually defined those media as parodies.

According to the definition of parody explored in the paragraphs above, however, similarity is not the chief defining characteristic of parody. Parody appraises the value of an original work by presenting a slightly different representation of the original. Parody even criticizes or mocks an artwork by editing the original and adding different images or scenes. What defines an artwork as a parody is its difference from its original sources (Hutcheon 2000, p. 38). Based on this artistic understanding of parody, what Korean churches have produced are not necessarily parodies but possibly pastiches.

For the confirmation of the suspicion regarding the term of reference, therefore, it is necessary to closely inspect some examples of Korean evangelistic media based on the conceptual knowledge of parody and pastiche. When it comes to examining the "Church Style" music video, for example, it is clear that the pastor who produced this video clip appropriates almost every element of the "Gangnam Style" music video. From his appearance, the pastor faithfully copies the original music video: he

---

[21]  By comparing pastiche to parody, Hutcheon defines pastiche as imitation which consists of superficial images. This describes the intent-less copying nature of pastiche well. The link between the model and the pastiche is imperceptible so that it is impossible to acknowledge the meaning of the models from the images of pastiche (Hutcheon 2000, p. 38).

dresses up like Psy and dances like him. He also puts much work into acting like the original singer in the music video by imitating Psy's funny facial expressions and comical gestures. The remarkable synchronization of the song and his lip movements in the video clip is also a result of his hard work of imitation.

The clumsy similarity is also found in the lyrics of "Church Style." The pastor who directed this video clip also rewrote the lyrics of the song by replacing some words with technical Christian terms and sang the song by himself. The overall flow of the lyrics followed the original song, but some Christian words blurted out randomly. The mixture of the original and the rewritten lyrics might be one of the most significant elements of "Church Style," which reveals the fact that this one is an imitation of a popular song. The different and yet so similar lyrics provide fun to viewers, who are familiar with the "Gangnam Style" and who also know its lyrics well. Overall, the "Church Style" music video has various similar images, movements, and words that define the video as an imitation of the "Gangnam Style."

While the "Church Style" music video has elements similar enough to the original to signal to viewers that it is an imitation, "Church Style" has several features that are different from the original music video. The main and the minor characters' appearances are different. The pastor is slimmer than Psy, and the minor characters are much less polished than the ones in the "Gangnam Style." The places where the "Church Style" music video was created are different from the source as well. The imitation is filmed in a place which, of course, is never shown in the original: church. Since it is partially rewritten, moreover, the lyrics also contain differences. It uses terms which are typically used in church settings, such as prayer, the Bible, worship, faith, and Amen. Since those terms are simply inserted into the original lyrics, some sentences of the rewritten lyrics sound contrived. The lyrics of the imitation are loosely organized and even seem meaningless and random at points. Nevertheless, it is still true that those technical church terms create a gap between the imitated and the imitation (Kim 2013b, pp. 190–94).

When it comes to examining the differences between "Church Style" and "Gangnam Style," it is difficult to find any critical intentions in them. The pastor does not appraise the value of the specific popular songs or the quality of its music video through those differences. He also neither criticizes Psy's music, nor does he present any theological reviews on "Gangnam Style." He simply borrows. Since Psy released his song, numerous popular music critics have evaluated "Gangnam Style" from diverse perspectives. Some positively reviewed it as a sarcastic criticism on Korean capitalist culture represented by the word, "Gangnam." Others negatively criticized the sub-standard expressions of the music video (Yoon 2015, pp. 411–13). But one does not find any of these critical approaches to "Gangnam Style" in the differences created by the Christian imitation. There are neither appraisals nor criticisms. It revises the original music video to reveal the fact that it was designed by a Christian pastor, and as something to be used within Christian communities.

In fact, the deficient condition of the imitation is not only the issue of "Church Style" but also the common characteristics of most Korean churches' evangelistic imitations. The movie posters, placards, and the flyers that Korean churches produced by copying popular cultural contents for the sake of Christian evangelism share the insufficiency; there is more similarity to the originals than differences—or at least the differences do not bear a critical relationship to the original artwork. Most evangelistic media produced by Korean churches that attempts to copy popular culture do not form any purposeful relationship to the original authors' intentions. The repetition of the "critical differences" are absent in those imitations (Hutcheon 2000, p. 64). Instead, there are edited images that only stimulate the interest of the viewers. There are cut-and-paste images of pastors, edited catch phrases, and appropriated themes and designs. One cannot easily discover any meaning in these differences except the fact that they are noticeable and provoke a laugh.

By contrast, there is an example of a Christian evangelistic medium that "parodies" a popular TV show. It critically interprets the original material and delivers its message by focusing on its differences from the source. It is a postcard promoting an online Bible study program that the Baltimore Baptist

Church in Asheville, North Carolina, sent out. This postcard contains an image, which, at a glance, looks like the opening title sequence of the popular TV show *Modern Family*. The theme of the picture, the title, and even the font is similar to the original image so that one can easily misunderstand that it is sent by the production company of the TV show.

However, a closer look at the postcard reveals that it is an imitation of the title image because it is slightly differentiated from its source. The church erases the image of the homosexual couple, who are among the title roles of the TV show and fills up the postcard with the pictures of three heterosexual families. The church designed this postcard to promote its online lecture program which deals with contemporary family issues. By intentionally omitting the image of the gay couple, this church reveals its antigay orientation, and at the same time, it criticizes the all-inclusive message of the original TV show. Judging from the aesthetical aspect of this postcard, it places weight on its difference and also conveys a critical reinterpretation of the original material. Whatever one thinks about the church's intolerance toward homosexuality, this postcard is an example of a Christian evangelistic parody of popular culture.

Comparing to the imitated postcard created by the Baltimore Baptist Church, Korean churches create a hybridity which is barely related to the intention of the original artwork. Therefore, pastiche is the proper term of reference for the evangelistic media that Korean churches make. They do not create any new meanings by reinterpreting or reconstructing the preformed images and texts. Rather, they share a "pure and simple" purpose: imitation (Park 2006, p. 260). They are "blank" in that they only fulfill the superficial meaning of parody (Jameson 1998, p. 5). They are "depthless" for they do not deliver any messages through their presentations, which are created by blending the existing and new images (Jameson 1992, p. 17). As a result, they are "schizophrenic" because they break the link between the imitated and the imitation so that no viewers can guess the meaning of the newly created images based on their knowledge of the old images (Jameson 1998, p. 34).

## 5. A Practical Theological Reflection on the Evangelistic Pastiches

From an aesthetical perspective, of course, the fact that those Korean evangelistic media are not parodies but pastiches of popular culture are not controversial. It is true that delivering Christian messages and promoting church events to people who are familiar with popular culture by utilizing famous music, movie, video clips, or TV programs can be effective. It can also demonstrate the church is trendy and "cool" because it is aware of what is going on in popular culture. Moreover, it can be a practical way of spreading the gospel to people of the 21st century who have diverse cultural preferences through the most universal, or the most enjoyable, way (Choi 2009, pp. 377–78). In fact, pastiche is an often-used method for marketing and advertising. Considerable numbers of ad-makers create advertisements with images that most people are familiar with. Popular images disarm people and make them relaxed so that they effectively attract the attention of potential customers. Thus, pastiche is possibly an efficient method for practicing Christian evangelism, solely from a pragmatic perspective.

However, it can be controversial if one approaches the evangelistic pastiches from the theological perspective. In his book, *Evangelism after Christendom: The Theology and Practice of Christian Witness*, Bryan Stone defines Christian evangelism as a practice, relying on Alasdair MacIntyre's concept of practice in *After Virtue*. According to Stone, this practice is aimed at goods internal to the practice itself, and the excellence of a practice is only judged by how well it fulfills its intrinsic values. The internal goods of a practice are determined by the community that cultivates the virtues requisite to the practice and the narratives that motivate the fulfillment and provide its aim (Stone 2007, pp. 31–32). It is often tempting to seek external goods, but those can never be the measure of a practice performed well. Based on this understanding of a practice, Stone affirms that Christian evangelism is a practice of a people whom God elects and forms as witnesses of and invitations to the peaceable reign of God (Stone 2007, p. 39).

Christian witness is embedded in how the people of God are related to the world. In the relationship, Christians bear witness in the very form of their social existence to a new reality in which

they exercise the peaceful and holy virtues of God's reign in their everyday life. Christian evangelism is subversive because it demands social changes and communal actions which eventually reveal the reign of God in this world (Stone 2007, p. 78). It is also invitational in that those evangelistic practices publicly herald the distinctive reality that is open to all creation.

The major point of Stone's theological understanding of evangelism is that the ultimate purpose of evangelism is the faithful, social embodiment of the gospel of God's reign. By practicing the virtues of the reign of God in the world throughout their lives, Christians invite non-believers into a new, distinctive reality. The result of evangelism is evaluated not by the numerical growth of new members of a church but by the expansion of the influence of the gospel in the world (Stone 2007, pp. 271–72). Assuming that every practice of evangelism can be an evangelistic medium, therefore, Christians should carefully consider the way they embody the gospel because it is directly related to the faithfulness of the practice of evangelism. While Stone does not use the language of media, for him, the messages proclaimed are embedded within the evangelists' living witness to the reign of God (Stone 2007, pp. 252–53).

Employing the insight from the theories of evangelism explored above, therefore, evangelistic media are fundamentally meant to be visible testimonies that socially embody the gospel about the peaceable reign of God. In addition, evangelistic media should illustrate how Christians faithfully practice the virtues of God's reign in their daily lives. If that is true, however, it is doubtful that the evangelistic pastiches created by Korean churches can be media that adequately communicate those significant messages. Insofar as those evangelistic tools are hybrids of popular cultural fun and superficial images of Christianity, what they deliver cannot be a substantive, much less comprehensive, witness to God or God's reign and its realization.

When Korean churches design the pastiches of popular culture as a means of evangelism, they simply hybridize popular cultural fun and superficial descriptions of the Christian gospel. The result is nothing but another kind of simulated images. These new evangelistic images neither embody the original messages of the popular cultural artists nor authentic messages about Christianity. What these media communicate instead is ultimately empty descriptions of the Christian gospel or Christian communities. They offer only the hyper-reality of Christianity. It seems barely possible to imagine the divinity living and working in human history from the evangelistic pastiches. Also, it is less reasonable to consider the messages of those evangelistic media as living witnesses of God. The gospel vanishes beyond the surfaces of those media and only the images of Christianity stimulating the peripheral nerves with popular cultural fun remain.

Like other artistic, nonreligious pastiches, therefore, Korean churches' pastiches are products of the technique of appropriation that solely aims at copying and editing randomly chosen images (Jameson 1992, p. 17). Pastiches are irrelevant to the originals or to reality, and therefore, the viewers of the pastiches cannot even begin to imagine the reality to which they might point. But of course, they don't point at all. Whether it is intentional or not, pastiches hide and efface the reality and only present hyper-real images of Christianity (Lane 2000, pp. 90–91).

Of course, the creators of the evangelistic pastiches of popular culture may refute the aforementioned evaluation by affirming that their media also communicate the gospel. However, based on the foregoing analysis of pastiche, it is not sure what the gospel is about. When it comes to considering the messages about the reign of God as good news, it is difficult to judge from a few church terms which are intermittently included among the images and video clips as also being the gospel. Even if one summarizes the entire life and work of Jesus Christ in the evangelistic pastiches, it will not make any difference. Pastiches only manifest blank images which feign the good news. The messages of the evangelistic pastiches are irrelevant to the gospel of the peaceable reality of God's reign. Delivering the gospel through imitations of popular culture which are the strategic media of postmodern consumerism result in the implosion of the fundamental messages of Christianity. The imploded gospel is not the content that Christians witness through the ministry of evangelism.

It is only a feigned Christian message represented by trendy and interesting images. It is, in short, a hyper-gospel.

As the pastor who directed the "Church Style" music video answered in an interview, one of the messages that the evangelistic pastiches of popular culture possibly deliver is about Christian communities' openness to the world (Yoo 2014). That message emphasizes the fact that Korean churches are culturally inclusive enough to utilize popular culture within their communities. The video clip of a pastor dancing to a popular song and the bible verses photo-shopped to famous movie posters possibly witness the cultural openness of Korean churches. In addition, the evangelistic media give an impression that the church is an entertaining community. When Korean churches create the evangelistic pastiches, they intentionally choose cultural contents that gain popular success. By adding fun elements to those contents, they produce evangelistic materials that everyone who knows the original works can enjoy. Therefore, when non-Christians see those materials, they may think that the people who design those contents are witty. It is not certain whether Korean churches plan to introduce themselves as amusing communities, but it is highly possible that the viewers of the pastiches may think churches are enjoyable.

However, the cultural openness and the fun-to-be-ness are not the internal goods (to use Stone's and MacIntyre's language) that Christians should realize through the practice of evangelism. They have nothing to do with the distinctively subversive characteristics of the reign of God (Stone 2007, p. 45). The evangelistic pastiches simply implant the afterimages of the ridiculously edited versions of pastors, churches, and the Bible. The fun-oriented hyper-gospel images replace the concrete exemplifications of political, economic, and social practices by which Christians bear witness to God's peaceable reign. Similarly to the icons that got rid of the divinity and made people unaware of the absence of the deity, the evangelistic pastiches of popular culture eventually efface the essential meaning and message of Christianity, which should be offered evangelistically to the world.

## 6. Conclusions

The argument above showed that it was difficult to find the essential differences that Korean churches intentionally added in the evangelistic appropriations of popular culture. I supposed that it was because the primary purpose of Korean churches' evangelistic imitation was practically utilizing popular culture to attract people's attention. However, the pragmatic approach to popular cultural materials made Korean churches overlook the aesthetical meaning of "parody." In that those imitations of popular culture merely presented random editions of famous images or video clips, I hypothesized that they are closer to pastiche than parody.

Those edited images of evangelistic pastiches deliver messages which are irrelevant to the meanings of their sources. Korean churches try to practice Christian evangelism by using a meaningless mixture of Christian and non-Christian cultural media. While people are fascinated by those unreal images of Christianity, evangelistic pastiches possibly mask the existence of the kingdom of God in the world. Therefore, I may conclude that the evangelistic pastiches may not witness the kingdom of God which is real and which is also practiced by Christians in the world.

However, the cultural hybridity that Korean churches create by imitating popular culture is still significant theologically in that it suggests the possibility that Christian evangelism and popular culture might be connected. The issue is how to build up the engagement with popular culture not in superficial and pragmatic ways but in critical and constructive ways. If one can connect them in a more robust way than the mere practice of copying the content of popular culture, Christians can effectively and faithfully communicate the gospel through the hybridity.

It is possible to imagine a relationship of Christianity and popular culture established through parody. Christians' approach to popular cultural elements can be understood as a parodic appropriation because Christians utilize the same kind of cultural materials in a different manner based on their understanding of the gospel. Moreover, a new use of popular cultural elements can be a witness of Christian faith. The usage of cultural materials within popular culture is basically consumerist and

mass-centered. The cultural forms of popular culture are also "vulgar" in that they reflect everyday life and experience (Pattison 1987, pp. 8–9). However, Christians utilize the cultural materials shared with popular culture in a different manner. The difference created by a parodic relationship with popular culture can be a means of Christian evangelism which proclaims a new, creative way of human life.

As an example, let us go back to the evangelistic postcard that the Baltimore Baptist Church made by parodying the famous American TV show, *Modern Family*. The evangelistic postcard is a good example of how Christianity and popular culture can be hybridized culturally in the form of parody. The church designed this postcard to invite people to its new online program which would deal with contemporary family issues. By intentionally replacing the image of the gay couple, which was shown in the opening title sequence of the TV show, with a non-gay couple, the church clearly presented its concerns about contemporary society and gave witness to its beliefs about the proper constitution of families. Although its interpretation of the gospel and the way it presented what it believed is controversial, the parodied medium clearly showed the church's critical approach to the contents of the original TV show, its understanding of the issues facing contemporary families, and its religious interpretation of those issues.

Whatever one thinks about the church's controversial position on the subject of homosexuality, the postcard affords us a glimpse of what an evangelistic parody that hybridizes Christianity and popular culture might look like. It appropriates cultural elements, such as images and themes, from popular culture. However, it adds a Christian interpretation that critically responds to contemporary issues in light of what they believe. The creative distance between the original content of popular culture and the evangelistic medium arouses interest for the viewers of the parody, and also delivers its message.

Insofar as Christians approach the popularity reflected of the contents of popular culture in a parodic way, the hybridity of Christianity and popular culture is evangelistic. It invites persons into the visible reality of the reign of God in which the "popular" concerns of humanity can find new, creative approaches through which people can imagine different ways of life. In this regard, evangelistic parodies are different from strategic evangelistic media that aim only at bringing people into the church and from the depthless utilizations of the entertaining contents of popular culture that simply stimulate people's interest. The evangelistic parody then can be a robust, not sensational, medium that reflects the faithful embodiment of the subversive and radical virtues of God's reign. It also presents a different interpretation of the meaning of everyday life that Christians and non-Christians experience in common.

**Funding:** This research received no external funding.

**Conflicts of Interest:** The author declares no conflict of interest.

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
