# Peer review of "The Appropriation of Popular Culture: A Sensational Way of Practicing Evangelism of Korean Churches"

_religions, doi:10.3390/rel10110592_

Round 1
Reviewer 1 Report
The author of this article has made substantive improvements since the first submission. Notably, the author has strengthened the explanation of why it might be significant that Korean churches employ pastiche rather than parody in making use of popular culture products. That is a big improvement. The newly expanded section makes a rather striking claim that by virtue of the definition/process of parody, churches that employ it are more closely expressing the nature of the Christian gospel, especially as a subversive force. It could be made a bit clearer whether this is true for any act of parody, or if the nature of the source being parodied makes a difference. It seems as though the author is claiming that it is the process of parody itself (rather than parody of any particular source) that somehow more honestly participates in the evangelistic mission. If so, adding a sentence or two to underscore that would help. Alternatively, if the actual source being parodied is important, it might be handy to provide an example(s). Perhaps returning to the postcard example might work?
Stylistically, the explanations are otherwise clear, though, as before, there is still a tendency to restate what has already been stated more than is absolutely necessary. Careful proofreading would be good. Specific lines with typos/glitches that I noticed (though there could be others) include:
line 33 better wording would be: "with which they are familiar"
line 105 pre-existing
footnote 5 the adjective Modernist or modernist seems needed rather than modernism
line 241 art works
line 615 than (rather than that)
line 625 drop "a" to march plural "ways"
Author Response
Point 1. The author of this article has made substantive improvements since the first submission. Notably, the author has strengthened the explanation of why it might be significant that Korean churches employ pastiche rather than parody in making use of popular culture products. That is a big improvement. The newly expanded section makes a rather striking claim that by virtue of the definition/process of parody, churches that employ it are more closely expressing the nature of the Christian gospel, especially as a subversive force. It could be made a bit clearer whether this is true for any act of parody, or if the nature of the source being parodied makes a difference. It seems as though the author is claiming that it is the process of parody itself (rather than parody of any particular source) that somehow more honestly participates in the evangelistic mission. If so, adding a sentence or two to underscore that would help. Alternatively, if the actual source being parodied is important, it might be handy to provide an example(s). Perhaps returning to the postcard example might work?
Response: First of all, I really appreciate your comments including the previous ones. With your help, I can improve the quality of my manuscript. Following your comments, I have strengthened the conclusion by bringing the Modern Family postcard back. I underscored the parodic relationship that the postcard made and also emphasized the practical value that the postcard showed. I think the conclusion would present a more concrete example of practicing Christian evangelism in relation to popular culture.
Point 2. Stylistically, the explanations are otherwise clear, though, as before, there is still a tendency to restate what has already been stated more than is absolutely necessary. Careful proofreading would be good. Specific lines with typos/glitches that I noticed (though there could be others) include:
line 33 better wording would be: "with which they are familiar"
line 105 pre-existing
footnote 5 the adjective Modernist or modernist seems needed rather than modernism
line 241 art works
line 615 than (rather than that)
line 625 drop "a" to march plural "ways“
Response: I have corrected incorrect expressions following your guidance. I have also proofread my article carefully and focused to shorten the paragraphs by reducing restated sentences. I found a considerable numbers of sentence which were not necessary so that I eliminated. I think it would be better for readers to follow the stream of the logic of my article without any stagnation.
Again, I really appreciate your careful review. I hope you enjoy the re-revised version of my article.
Sincerely,
Author
Reviewer 2 Report
This article needs substantial restructuring and revision before it could be accepted for publication. The topic and introduction read as interesting but from lines 50 to 425 there are long explanations of basic European literary and art-history concepts that have little to do with the topics of religion or evangelism. Readers need instead to understand what is judged to be genuinely 'religious' (inspiring, moving) art in a Korean context (or indeed in a European context - is there agreement on what constitutes the spiritual in art? before we can evaluate whether the hybridisation of evangelical effort with popular culture results in image-making that exhibits 'insufficiency' or is 'blank parody' or perhaps, is 'subversive' in a way that exhibits the presence of God [even in popular culture], although I realise the latter conclusion is unlikely. (By contrast Hendershot (2004) writes about the vibrancy of evangelical material culture in the United States as Christians set up an alternative cultural economy). There are likely to be more contemporary articles on related topics and in general the use of more contemporary reference sources would help.
The distinction between 'parody' and 'pastiche' is ultimately useful but we first need to understand the etymology of the word 'parody' in Korea specifically -in what contexts is it used, by whom, and with what meaning? This meandering, blandly international section (from 50-425) could be cut down to 2 or 3 pages to get to the more relevant and specific sections 4 and 5 more quickly. The article in general could possibly be re-focused around examples from different types of Korean religion so we can understand if this evangelical re-purposing of popular culture, in a pastiche-mode, is unique. For example, in my experience of having visited it, the former home and museum of the late Rev Moon employs an exuberant, pastiche-mode, visual aesthetic which suggests that there are precedents for, and perhaps an underlying disposition towards, the employment of popular culture imagery and tropes in the religious sphere in Korea.
Author Response
Point 1. This article needs substantial restructuring and revision before it could be accepted for publication. The topic and introduction read as interesting but from lines 50 to 425 there are long explanations of basic European literary and art-history concepts that have little to do with the topics of religion or evangelism. Readers need instead to understand what is judged to be genuinely 'religious' (inspiring, moving) art in a Korean context (or indeed in a European context - is there agreement on what constitutes the spiritual in art? before we can evaluate whether the hybridisation of evangelical effort with popular culture results in image-making that exhibits 'insufficiency' or is 'blank parody' or perhaps, is 'subversive' in a way that exhibits the presence of God [even in popular culture], although I realise the latter conclusion is unlikely. (By contrast Hendershot (2004) writes about the vibrancy of evangelical material culture in the United States as Christians set up an alternative cultural economy). There are likely to be more contemporary articles on related topics and in general the use of more contemporary reference sources would help.
Response: First of all, I appreciate your detailed comments. Following your guidance, I have eliminated a considerable portion of the lengthy explanation part including the modernist understanding of appropriation. I have also revised the latter part of my article so that readers can have enough understanding about the relationship between Christian evangelism and arts.
Point 2. The distinction between 'parody' and 'pastiche' is ultimately useful but we first need to understand the etymology of the word 'parody' in Korea specifically -in what contexts is it used, by whom, and with what meaning? This meandering, blandly international section (from 50-425) could be cut down to 2 or 3 pages to get to the more relevant and specific sections 4 and 5 more quickly. The article in general could possibly be re-focused around examples from different types of Korean religion so we can understand if this evangelical re-purposing of popular culture, in a pastiche-mode, is unique. For example, in my experience of having visited it, the former home and museum of the late Rev Moon employs an exuberant, pastiche-mode, visual aesthetic which suggests that there are precedents for, and perhaps an underlying disposition towards, the employment of popular culture imagery and tropes in the religious sphere in Korea.
Response: I was personally pleased to know that you have experienced Korean context. And, I also agree with your comments in that it would be helpful to widen the perspective of my article to other Korean religions. I have tried to find examples that other Korean religious communities produced any media utilizing popular cultural contents to inform their religious traditions. However, my researching skill was not enough to find any proper examples. Instead, I have focused on the evangelistic practice of Korean Christianity and also tried to suggest a creative way of producing evangelical, subversive Christian media at the end of my article. I am sorry that I could not engage the comparison between Korean Christian evangelistic media to other religious media in this article, but it will be a good topic for my next article.
Thank you for giving me an academic inspiration. I hope you enjoy the revised version of my manuscript.
Sincerely,
Author
Round 2
Reviewer 2 Report
Dear Author, Thank you for your respectful response to my first set of comments and for the changes you have already made. I am pleased if I was able to give you an idea for future research. I have read this second version very carefully and made many notes, trying to work out exactly what my issues with the article are, noting that there are still significant issues remaining, but also noting that you are clearly committed to making a fine-grained argument, which is a strong positive feature of the article.
As soon as I finished that I did what I should have done first - went online - and found what I am sure must be your Ph.D thesis, 'Parody and Pastiche in the use of popular culture in the evangelical practices of Korean Churches', finished last year. Reading both the Table of Contents and introductory pages of your thesis, which look to be of a good level of quality I feel I can be more direct with you now.
It is immediately clear that the Ph.D is in the discipline of theology and is the work of a Christian author. This is not clear in the article, until around line 540 where the argument suddenly takes a theological and moral turn, asserting that Christian communications ought to 'bear witness to God" and that you are of the opinion that the depthless style of these appropriations of popular culture cannot do that job with integrity.
Because you do not state your position until that late in the article myself, as a reader, felt disorientated and uncomfortable for most of its length. I did not really understand what the point of what you were arguing was, nor its significance for a non-Korean audience - I felt as if you were talking 'inwards' to a community who might have a better idea of the applicability of your argument. By contrast, you were not giving the rest of us enough specific information about contemporary Korean evangelical Christianity - its history, characteristics, goals and challenges (but I see there is a section on the history of Korean Christianity in your thesis) so that we could see for ourselves why the distinction between parody and pastiche is important, although the rewritten version of your conclusion is stronger on that point
Instead, for most of the article you make it seem as if these pastiche pastors live in a bland globalized world where, apart from remaking a video or poster or two, ministry and community are probably conducted in English and that many cultural assumptions and concerns are also indistinguishable from those of American evangelicalism for instance. (Is that correct? you did not respond to my question about what the word for 'parody' is in Korean and how/why it came to be so widely used as a description. It is essential you discuss, not just assert, this point. Why are Korean Christians so invested in this particular concept?).
In my opinion, you should make one more revision of the beginning of this article - being clear about your personal stance and committed style of scholarship, briefly writing about the history etc of Korean Christianity, and being more overt about your Research Question, Approach/Method and Hypothesis (I see the second mentioned in your thesis.) You could also usefully make more links back and forwards in your argument as you progress through the long, careful expositions of your understandings of parody and pastiche, so that we can keep alive the significance of the distinctions you are making.
I apologize for the trouble I am causing you but I would not be comfortable recommending publication until the article is more 'open' about its own assumptions and values.
Author Response
Dear Reviewer,
I deeply appreciate your time and endeavor that you spent for reviewing my article. I also thank you for your thoughtful comments which were really helpful for me to improve the quality of my writing.
First of all, I have revised the introduction following your advice. I have added several paragraphs which would help readers to understand the background of the contemporary Korean Christianity. In addition, I have tried to make my academic stance clear so that people who may read my article would notice that this is a practical theological article.
To answer your question about "parody," there is no Korean word which can replace "parody." Although it is a foreign term of reference, we, Koreans use is as it is. I think this fact may help you to understand the context that I am dealing with in my article. Those pastors are serving churches in Korean context, while they use the term, "parody." I also have explained the reason why using parody became so important in Korean Christian context in the introduction. I think your point was really accurate.
Again, thank you for your comments and I hope you enjoy this version of my article.
Peace of the Lord,
Min
Round 3
Reviewer 2 Report
Thank you for making those additional changes. The paper is now more tightly focused on its subject and its attitude to, and investment in, the topics of evangelical parody and pastiche is more openly acknowledged.
I am now of the opinion that the paper can be published and find it's audience.